# Deconstructing the Supermarket: Systematic Ingredient Disaggregation and the Association between Ingredient Usage and Product Health Indicators for 24,229 Australian Foods and Beverages

**DOI:** 10.3390/nu13061882

**Published:** 2021-05-31

**Authors:** Allison Gaines, Maria Shahid, Liping Huang, Tazman Davies, Fraser Taylor, Jason HY Wu, Bruce Neal

**Affiliations:** 1Department of Epidemiology and Biostatistics, Faculty of Medicine, School of Public Health, Imperial College London, London SW7 2AZ, UK; bneal@georgeinstitute.org.au; 2The George Institute for Global Health, University of New South Wales, Sydney, NSW 2042, Australia; mshahid@georgeinstitute.org.au (M.S.); hliping@georgeinstitute.org.au (L.H.); tdavies@georgeinstitute.org.au (T.D.); ftaylor@georgeinstitute.org.au (F.T.); jwu1@georgeinstitute.org.au (J.H.W.)

**Keywords:** ingredients, food labelling, health star rating, nutrient profiling, ultra-processing, packaged foods

## Abstract

Unhealthy diets are underpinned by the over-consumption of packaged products. Data describing the ingredient composition of these products is limited. We sought to define the ingredients used in Australian packaged foods and beverages and assess associations between the number of ingredients and existing health indicators. Statements of ingredients were disaggregated, creating separate fields for each ingredient and sub-ingredient. Ingredients were categorised and the average number of ingredients per product was calculated. Associations between number of ingredients and both the nutrient-based Health Star Rating (HSR) and the NOVA level-of-processing classification were assessed. A total of 24,229 products, listing 233,113 ingredients, were included. Products had between 1 and 62 ingredients (median (Interquartile range (IQR)): 8 (3–14)). We identified 915 unique ingredients, which we organised into 17 major and 138 minor categories. ‘Additives’ were contained in the largest proportion of products (64.6%, (15,652/24,229)). The median number of ingredients per product was significantly lower in products with the optimum 5-star HSR (when compared to all other HSR score groups, *p*-value < 0.001) and significantly higher in products classified as ultra-processed (when compared to all other NOVA classification groups, *p*-value < 0.001). There is a strong relationship between the number of ingredients in a product and indicators of nutritional quality and level of processing.

## 1. Introduction

Much ill health globally is caused by poor dietary choices [1]. Unhealthy diets are, in turn, underpinned by a food system that tends to promote over-consumption of unhealthy packaged foods and beverages [2,3]. Often these packaged products are high in salt, added sugar and harmful fats which are direct causes of obesity, diabetes and other chronic health conditions [4].

It can be difficult for consumers to pick between more and less healthy packaged food and beverage products. To provide consumers with point-of-sale information that identifies better choices, governments and other organisations are introducing front-of-pack labels. For example, the Health Star Rating (HSR) system [5] developed by the Australian government assigns each food between half a star and five stars using a nutrient profiling algorithm. The NOVA classification system [6] is another method of distinguishing better-for-you options by defining food and beverage products according to the level of processing involved in manufacturing [7]. A number of studies assessing ultra-processed food intake have found an association between greater consumption of ultra-processed food and increased risks of obesity, cardiovascular disease and mortality [8,9,10].

Direct assessment of ingredients may be another way to evaluate the healthiness of packaged food products. Ingredient labelling standards and food composition tables in Australia are governed by Food Standards Australia New Zealand (FSANZ). Ingredient labelling is designed to provide consumers with transparency into product composition. FSANZ legislation requires a statement of ingredients to be arranged in descending order by ingoing weight, allowing consumers to discern the primary components of a product’s recipe [11]. In addition, explicit percentage labelling is required for ‘characterising ingredients’ (e.g., strawberry in a strawberry yoghurt) [12]. Food composition tables provide generic reference data describing nutrient content in common foods and can help assess dietary intake [13,14], but do not provide specific information about individual packaged food products.

The numbers and types of ingredients in the packaged food and beverage supply may provide a further means of assessing healthiness [15]. Novel, ingredient-centric work may contribute to improving Australian food composition databases as well as enable studies on ingredient interactions in composite foods [16,17], particularly by better defining ingredient data available on packaged products.

In this study, we sought to define the list of ingredients contained in packaged food and beverage products marketed at major grocery retail outlets in Australia in 2019. We categorised the ingredients identified and explored the associations between the number of ingredients used per product and indicators of product healthiness.

## 2. Materials and Methods

This was a cross-sectional examination of packaged food and beverage products in the Australian food supply carried out using FoodSwitch, a database developed and operated by the George Institute for Global Health.

### 2.1. Food and Beverage Product Database

The 2019 FoodSwitch Annual Database, the dataset used in this study, was generated from in-store surveys carried out in Australia at five large supermarkets (IGA, ALDI, Woolworths, Coles and Harris Farm) between August and November 2019 [18]. The dataset represents the majority of supermarket food and beverage products purchased by Australian households [19,20].

The FoodSwitch database holds up to 400 data fields for each included product. Data are obtained direct from packaging (e.g., product name, package size, ingredients, nutritional claims and nutrient content) or derived from data on the packaging (e.g., HSR and NOVA classification). Products held in the FoodSwitch database are classified into a hierarchical categorisation evolved from a system initially developed by the Global Food Monitoring Group [19]. The categorisation system was designed to enable the description and tracking of the nutritional composition of products in groupings meaningful to diverse stakeholders including academic researchers, the food industry, government and consumers [21,22]. There are 18 major food and beverage categories but products are further allocated into hundreds of finer subcategories.

Products were excluded from this analysis if they carried multiple nutrition information panels (e.g., variety packs) or were not in a relevant major food category (i.e., ‘Alcohol’, ‘Vitamins and supplements’ or ‘Unable to be categorised’). Extensive data cleaning was undertaken but some products were also excluded for having an incomplete or non-standard statement of ingredients.

### 2.2. Identification of Ingredients

Ingredients were identified from the images of the product label held within the FoodSwitch system and were obtained from either the statement of ingredients or the product name.

For products with a statement of ingredients, data were transcribed into the database as delimited text strings reflecting the form of information provided on pack. The text strings were disaggregated using an automated process creating separate fields for each ingredient and extracting compound ingredient information when present (Table A1 in Appendix B).

Compound ingredients are ingredients made up of two or more sub-ingredients [11]. Sub-ingredients require declaration if they constitute more than 5% of the final food or beverage product or contain a known allergen [23]. There were three forms of compound ingredient presentation: (1) those for which the listed sub-ingredients together made up the entire compound ingredient (complete sub-ingredients), for example, where fortified wheat flour sub-ingredients were listed as wheat flour, thiamine and folic acid; (2) those for which the listed sub-ingredients comprised only the additives or fortificants (incomplete sub-ingredients), for example, where fortified wheat flour sub-ingredients were listed as just thiamine and folic acid; and (3) those for which sub-ingredients were not listed.

Extensive cleaning of the disaggregated ingredient data was done using a semi-automated process (Figure A1). Key elements of the process were the removal of symbols (e.g., reduced-fat milk became reduced fat milk), the correction of spelling errors (e.g., watermellon became watermelon) and the alignment of plurality (e.g., raspberries became raspberry). For ingredients with various common names, a manual process to align terminology was undertaken (e.g., garbanzo bean became chickpea). Descriptive terms were removed from ingredient names (e.g., natural freshly squeezed orange juice became orange juice). When a term relating to an agricultural technique was identified (e.g., organic or free-range) it was removed and stored separately. Approximately 20% of ingredient extractions were manually validated.

For products with ingredients in the product name, a manual ingredient identification and extraction process was performed. The FSANZ Ingredient Labelling of Food standard precludes products from requiring a statement of ingredients in three cases: (1) where the product is labelled with the name of the food and the name of the food includes all ingredients, (2) where the product is water in a packaged form and (3) where the product is contained in a small package (surface area less than 100 cm^2^) [11]. For products without a statement of ingredients but whose ingredient information could potentially be determined from the product name, a text-matching algorithm was used to extract ingredient information. For example, the ingredient ‘blueberry’ was extracted from the product name ‘Fresh Blueberries’. In some cases, multiple ingredients were extracted from the product name, for example, the ingredients ‘pork mince’ and ‘veal mince’ from the product ‘Pork & Veal Mince’.

### 2.3. Categorisation of Ingredients

Ingredients were categorised into a single, hierarchical structure with each ingredient assigned a major and minor category. Category naming conventions and the category structure were determined by discussion amongst the authors and guided by the terminology used by two specialty ingredient companies: Cargill and ADM [24,25]. Minor categories for ‘Sweeteners’ were verified against prior publications on sugar types contained in packaged food products [26]. A ‘Compound ingredients’ category was created to capture any main ingredients which, by name, were clearly composed of multiple sub-ingredients (e.g., chocolate).

Ingredients were flagged depending upon whether they constituted a substantive part of any product (substantive ingredients) or were always a small component of the products (non-substantive ingredients). Non-substantive ingredients were defined as ingredients that are usually less than 2% of a product formula, such as ingredients with maximum legal usage rates such as certain preservatives, vitamins and minerals; or ingredients that are always used at low levels to provide functional benefits such as gums and other stabilisers [27]. Colours, flavours and most additives were considered non-substantive ingredients.

E numbers were cleaned and listed under the ingredient’s common designation, such as the chemical name or additive type. For example, E100 through E199 were listed under the ingredient name colour. Similarly, rare substantive ingredients were grouped under a more common ingredient name where possible (e.g., za’atar was listed as dried herb).

### 2.4. Analysis

The total number of ingredients was calculated by tallying the number of ingredients at the most granular disaggregation level after the removal of duplicates (e.g., ‘canola oil, butter (cream, water, salt), eggs, white vinegar, salt’ contains six ingredients since salt was listed twice). Ingredients that were different prior to cleaning but the same after cleaning (e.g., E101 and E102 both classified as colour) were treated as distinct ingredients. Compound ingredients that contained an incomplete or missing sub-ingredient list were considered singular ingredients.

Descriptive statistics were presented for the average number of ingredients per product overall and for main food categories. We assessed the association between the number of ingredients per product and HSR, as well as NOVA classification. Differences in the number of ingredients across HSR scores and NOVA classifications were evaluated using a Kruskal–Wallis H test with post-hoc Dunn pairwise analyses. A subsidiary analysis was done using ingredient numbers based on main ingredients alone (i.e., excluding sub-ingredients). All data preparation and statistical analyses were conducted using Stata IC 15 (StataCorp) and Excel. A two-sided *p*-value < 0.05 was considered statistically significant. Ethical approval was not required for this study.

## 3. Results

A total of 27,366 food products were included in the Australian 2019 FoodSwitch Annual Database. Of these, 655 were in excluded product categories or were multi-pack products with more than one nutrition information panel, and 441 had no available means of identifying an ingredient list. A further 2041 products had a disorganised statement of ingredients format so disaggregation of the ingredients was not possible (Figure A2). This left 24,229 products for analysis. Of these, 22,493 (92.8%) displayed the statement of ingredients on pack and 1736 (7.2%) required extraction of ingredients from the product name. The majority of products requiring ingredient extraction from the product name had one ingredient (*n* = 1732) and the remaining four products had two ingredients.

### 3.1. Identified Ingredients

The 24,229 products listed a total of 233,113 ingredients, producing a list of 31,934 unique ingredient names. Semi-automated cleaning reduced this number to 915 distinct ingredients (Figure A3). The 915 distinct ingredients were categorised into 17 major categories (Table 1) and 138 minor categories (Appendix A in the Appendix A). Of the ingredients, 835 were identified as nutritionally substantive and 80 were identified as nutritionally non-substantive. The majority of unique ingredients belonged to the major category ‘Fruit and vegetables’ (*n* = 229), followed by ‘Compound ingredients’ (*n* = 128), ‘Additives’ (*n* = 86) and ‘Liquids’ (*n* = 72).

A total of 9283 (38.3%) products contained at least one compound ingredient with sub-ingredients and 4631 (19.1%) products contained at least one compound ingredient without sub-ingredients. In total, 21,420 compound ingredients were listed. Across all compound ingredient types, 10,650 (49.7%) contained a complete sub-ingredient list, 6139 (28.7%) contained an incomplete sub-ingredient list and 4631 (21.6%) did not contain sub-ingredients which could be disaggregated.

### 3.2. Most Frequently Used Ingredients

As seen in Table 1, at the major ingredient category level, ‘Additives’ were contained in the largest proportion of products (64.6%, (15,652/24,229)), followed by ‘Herbs and spices’ (56.8% (13,766/24,229)) and ‘Sweetener’ (51.3% (12,437/24,229)). A total of 44.5% (10,720/24,229) of products had at least one ingredient categorised under ‘Fruits and vegetables’ and only 10.6% (2579/24,229) had at least one ingredient under ‘Legumes’.

Across all included products, salt (11,182/24,229) and sugar (9872/24,229) were the most frequently used ingredients. ‘Additives’ (i.e., flavours, colours, emulsifiers, acidity regulators, preservatives and thickeners) represented 6 of the 10 most frequently used ingredients (Table 2).

### 3.3. Number of Ingredients Per Product across Product Categories

Products had between 1 and 62 ingredients, though about 90% of products had fewer than 20 ingredients and single-ingredient products made up 18.1% (4381/24,229) of the total (Figure 1). The median (interquartile range (IQR)) number of ingredients across all products was 8 (3–14) (Table 3). ‘Convenience foods’ had the greatest median number of ingredients per product (20 (14–27)), followed by ‘Foods for specific dietary use’ (12 (7–22)) and ‘Bread and bakery products’ (13 (9–19)) (Table 3).

Excluding sub-ingredients reduced the total count of ingredients across all products to 165,011. The range in number of ingredients was decreased to 1 to 42, and the median number of main ingredients across all products was decreased to 6 (2–10). The median number of ingredients was reduced across all major food categories and markedly for ‘Convenience foods’ (from 20 to 10).

### 3.4. Associations of Number of Ingredients with Healthiness and Level of Processing

The number of ingredients per product was lower in healthier products with higher HSR. Products with 4-star, 4.5-star or 5-star ratings had a significantly lower median ingredient count than all other HSR values (all *p*-values < 0.001). The highest HSR, a 5-star rating, had the lowest median number of ingredients (1 (1–4)). The association was, however, not linear, with numbers of ingredients across HSR values from 0.5 to 3.5 being approximately similar. A progressive decline in numbers of ingredients used per product was observed across HSR values from 3.5 to 5.0 (Figure 2A).

A similar pattern was seen across the four NOVA classification groups. Ultra-processed foods (NOVA group 4) used the highest number of ingredients per product (8 (3–14)) while minimally processed foods (NOVA group 1) used the lowest number of ingredients (1 (1–3)). The median number of ingredients per product was significantly higher in ultra-processed products (NOVA group 4) compared to all other NOVA groups (all *p*-values < 0.001); similarly, the median number of ingredients in processed products (NOVA group 3) was significantly higher compared to NOVA groups 1 and 2 (all *p*-values < 0.001). The association between number of ingredients per product and NOVA group was non-linear, with similarly low numbers of ingredients in both minimally processed foods (NOVA group 1) and processed culinary ingredients (NOVA group 2) (Figure 2B).

The difference in the number of ingredients used across HSR and NOVA groups was attenuated but the same pattern remained in analyses based on main ingredients alone.

## 4. Discussion

There are many tens of thousands of packaged food and beverage products available in Australia, but these products are comprised of a much smaller number of ingredients. Of these ingredients, the most commonly used are salt, sugar, flavours, colours and other additives which are widely considered to be indicative of unhealthy foods [28,29]. At the same time, a higher number of ingredients in a product is strongly associated both with lower (i.e., unhealthier) HSR values and NOVA classifications indicating greater degrees of food processing.

The association between the median number of ingredients and nutritional quality was clear, though only apparent for products at the healthier end of the HSR spectrum. At HSR below 3.5 stars, the relationship between the number of ingredients and healthiness plateaued. The reason for this is not clear. A more detailed investigation of the healthiness of individual ingredients in a product with respect to the product HSR was beyond the scope of this initial report but might provide further insights in the future.

Using this large and representative dataset, we ascertained that the median number of ingredients used in products was positively correlated with higher NOVA classifications for levels of processing. More extensive processing done to enhance palatability, convenience and large-scale manufacturing often results in end products comprised of many components that have little in common with the base ingredients. Complex food chemistry may be involved, requiring multiple different additives, albeit usually in small quantities. Extensive processing is also associated with products being less healthy [10] and this correlates well with the observed associations between HSR and number of ingredients using this comprehensive dataset.

The NOVA classification system defines ultra-processed foods as formulations made mostly or entirely from substances derived from foods and additives ‘typically with five or more and usually many ingredients’ [30]. Thus, as previously suggested, products classified as ultra-processed were expected to contain a larger number of ingredients [6,31]. However, this is not always the case. For example, rice cakes contain only one to two ingredients (rice and, in some cases, salt), yet undergo numerous heat treatments and product modifications to cook, puff and compress the ingredients into the final product. This raises the question of which factors, including chemical, biological and cultural, should be considered when defining foods as ultra-processed. In fact, terminology around food processing and usage in policy making is of growing concern [32,33]. Currently, there is no single, accepted standard for classifying a food product’s level of processing [34]. Our results support the need for a more comprehensive classification system as well as further work to understand how processing impacts diet-related research and policy.

The research also identified several inconsistencies in ingredient labelling that may mitigate the goal of providing consumers with transparency into product composition, a pattern which has also been recognised in an analysis of United States product ingredient labelling [35]. The vast range of terms used in ingredient lists as compared to the actual number of different ingredients almost certainly introduces errors in consumer interpretation and understanding. Likewise, the use of compound ingredients, with inconsistent requirements for reporting of sub-ingredients, is another point at which misinterpretation of product ingredients is likely. While the ‘Compound ingredients’ category had a large number of unique ingredients, the usage of each was small, and it would be relatively little work for the affected companies to disclose those sub-ingredients.

There is a clear opportunity for better standardisation of ingredient reporting to enhance consumer understanding, as well as nutrition research. For example, presently, ingredient naming can be done using common names or generic names—if a product includes pear, apple and peach, a valid statement of ingredients can use either the discrete list of all three ingredients or the generic ingredient fruit [11]. It is also currently permissible for an ingredient to be omitted if it makes up a small proportion of the final product and is not a common allergen, or is a substance used as a processing aid or product flavouring [11]. This may not be a substantial issue in many cases, but in some circumstances it may be important. For example, vitamin and mineral fortification of packaged products, as well as the use of additives, is of increasing interest to consumers and governments around the world [36], and only with full reporting of all ingredients will such data be generally available. There is a strengthening case for both better specification of the regulations governing ingredient reporting as well as their more rigorous enforcement.

The novel ingredient dataset developed for this study enables future research in food composition analysis. Understanding the interactions between food ingredients, particularly when ingredients are processed or cooked in different ways, is of increasing interest in nutrition and food chemistry research [37]. In addition, our enhanced ingredient-level data defining the packaged food and beverage supply can contribute to the new nutrition science concept, which combines biological, environmental and social sciences [38]. For example, our methods can be applied to incorporate ingredient list data into widely used databases to better analyse the packaged food and beverage supply (e.g., FoodEx and LanguaL). There is also potential to address sustainable diets using ingredient breakdowns [39].

A key strength of our study is the size of the dataset on which we have based our analyses. The FoodSwitch platform systematically collects product information from all the main retailers, and these products represent the majority of all packaged food and beverages sold in Australia each year. The extensively cleaned and concise final list of categorised ingredients provides for a unique new description of the main constituents of the Australian packaged food and beverage supply. Future work using ingredient descriptors such as ‘organic’ and ‘free-range’ will provide additional opportunities for research and greater insight into the health and environmental characteristics of Australian packaged foods. The large number of carefully defined data points also enabled us to reliably describe the numbers of ingredients in different food categories and how they relate to widely used indicators of healthiness and food processing.

There are also some limitations. For about one-in-ten products, it was not possible to obtain an analysable statement of ingredients, and this was likely a more pronounced issue for products with larger numbers of ingredients. While extensive work was done to standardise ingredient terminology, this work was mostly done by one author and there may be some misclassifications. Workload limitations did not allow us to fully categorise different E numbers, which is a growing area of interest in nutrition research and will be a future task for us [40].

## 5. Conclusions

The assessment of constituent ingredients provides a novel means of evaluating packaged food and beverage products. Our initial findings related to HSR and NOVA classifications suggest that further exploration of food products based upon their ingredients may provide new insights into their effects on both human health and planetary health.

## Figures and Tables

**Figure 1 nutrients-13-01882-f001:**
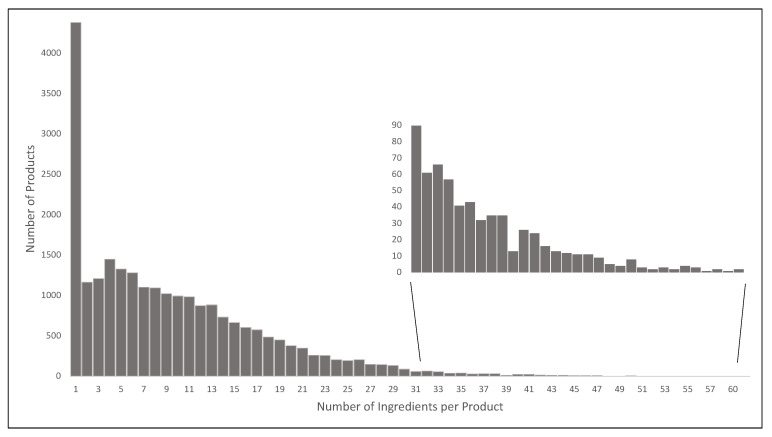
Distribution of the number of ingredients per product. The bar graph displays how many products contain the specified number of disaggregated ingredients in the statement of ingredients. The inset magnified graph displays the number of ingredients per product where product numbers are < 100.

**Figure 2 nutrients-13-01882-f002:**
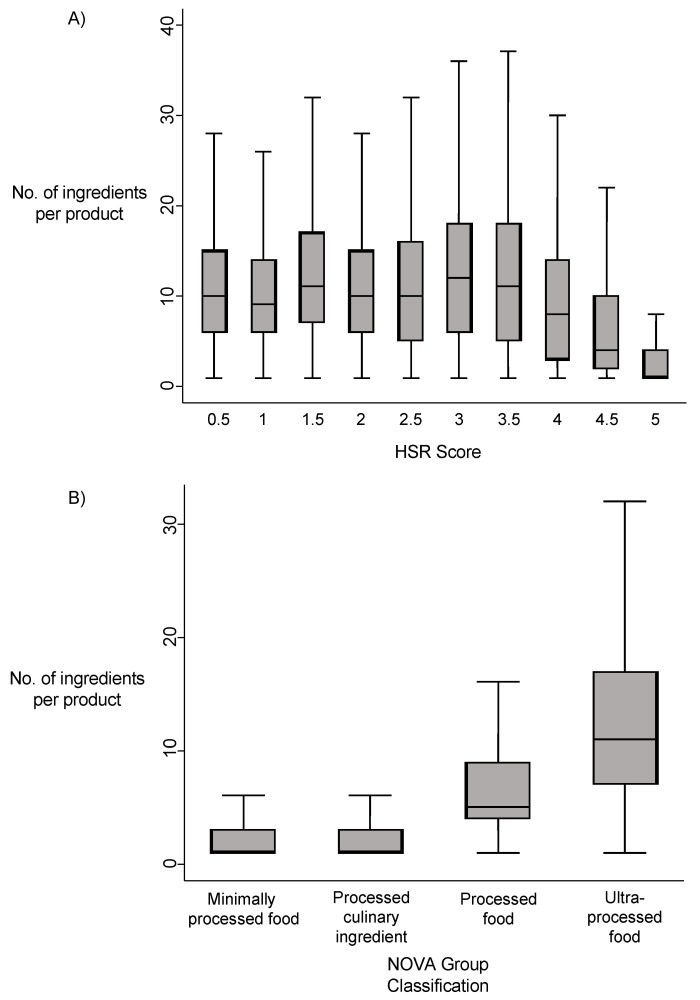
The association of the number of ingredients with (**A**) Health Star Rating and (**B**) NOVA classification. Box plot (**A**) shows the median and interquartile range (IQR) number of ingredients in products assigned different Health Star Rating (HSR) scores. The whiskers are defined as the lower quartile − 1.5 * IQR and the upper quartile + 1.5 * IQR. Outliers are excluded from the graph. Box plot (**B**) shows the median and interquartile range (IQR) number of ingredients in products assigned different NOVA classification groups. The whiskers are defined as the lower quartile − 1.5 * IQR and the upper quartile + 1.5 * IQR. Outliers are excluded from the graph.

**Table 1 nutrients-13-01882-t001:** Ingredient frequencies by major ingredient category.

Major Ingredient Category	No. of Unique Ingredients	Proportion of UniqueIngredients	No. of Non-Substantive Ingredients ^3^	Frequency across All Products ^1^	Proportion across All Products ^2^
Additives	86	9.4%	67	15,652	64.6%
Herbs and spices	35	3.8%		13,766	56.8%
Sweeteners	30	3.3%	8	12,437	51.3%
Liquids	72	7.9%		11,767	48.6%
Fruits and vegetables	229	25.0%		10,772	44.5%
Oils and fats	33	3.6%		9914	40.9%
Flours and starches	17	1.9%		7245	29.9%
Dairy	56	6.1%	1	7203	29.7%
Compound ingredients	128	14.0%	1	4980	20.6%
Baking ingredients	10	1.1%	1	4977	20.5%
Grains	48	5.2%		4059	16.8%
Nuts and seeds	31	3.4%		2809	11.6%
Meat	60	6.6%		2652	10.9%
Legumes	27	3.0%		2579	10.6%
Eggs	8	0.9%		1699	7.0%
Seafood	41	4.5%		1052	4.3%
Dietary fibres	4	0.4%	2	914	3.8%
	915	100%	80		

^1^ Frequency and ^2^ proportion across all ingredient lists describes the number/proportion of the 24,229 foods for which one or more of the ingredients in each major ingredient category was included in the ingredient list on pack. ^3^ Non-substantive ingredients were defined as ingredients usually less than 2% of a product formula, such as preservatives, vitamins and minerals, gums and other stabilisers.

**Table 2 nutrients-13-01882-t002:** The top 15 ingredients in the 2019 Australian packaged food and beverage supply.

Ingredient Rank ^1^	Ingredient	Ingredient Major Category	Substantive Ingredient	Frequency ^2^	Proportion^3^
1	Salt	Herbs and spices	N	11,182	46.15%
2	Sugar	Sweeteners	N	9872	40.74%
3	Water	Liquids	N	7971	32.90%
4	Flavour	Additives	Y	7533	31.09%
5	Colour	Additives	Y	4392	18.13%
6	Wheat flour	Flours and starches	N	3780	15.60%
7	Emulsifier	Additives	Y	3771	15.56%
8	Acidity regulator	Additives	Y	3124	12.89%
9	Vegetable oil	Oils and fats	N	3102	12.80%
10	Thickener	Additives	Y	3015	12.44%
11	Preservative	Additives	Y	3013	12.44%
12	Yeast	Baking	N	2763	11.40%
13	Milk powder	Dairy	N	2676	11.04%
14	Dried herbs	Herbs and spices	N	2470	10.19%
15	Vitamins	Additives	Y	1369	5.65%

^1^ Rank = ingredients are ranked according to how many of the ingredient lists they are included on out of the 24,229 included packaged food and beverage products. ^2^ The frequency is the number of products for which the ingredient is listed. ^3^ The proportion is the percent of the 24,229 total products evaluated.

**Table 3 nutrients-13-01882-t003:** The mean number and ranges of ingredients, overall and in major food categories.

	Total Products	Number of Ingredients						
		Mean ^1^	SD ^2^	Min	25%	50%	75%	Max
All	24,229	9.6	8.3	1	3	8	14	62
By food category:								
Convenience foods	1425	21.0	10.1	1	14	20	27	61
Foods for specific dietary use	2476	15.0	10.6	1	7	12	22	50
Bread and bakery products	2393	14.6	7.4	1	9	13	19	49
Meat and meat alternatives	726	11.9	9.6	1	1	11	18	53
Snack foods	1930	11.8	6.5	2	6	12	17	38
Sauces, dressings, spreads and dips	685	11.7	6.5	1	7	12	16	43
Confectionery	1351	10.8	6.0	1	7	9	13	62
Cereal and grain products	2556	9.2	8.7	1	1	6	15	58
Dairy	2990	8.3	6.7	1	4	6	11	56
Seafood and seafood products	1978	7.2	6.1	1	3	5	11	41
Non-alcoholic beverages	768	5.5	4.6	1	1	5	8	37
Fruit, vegetables, nuts and legumes	3923	4.2	4.4	1	1	3	6	50
Sugar, honey and related products	441	4.0	4.1	1	1	2	6	26
Edible oils and oil emulsions	498	3.4	3.9	1	1	1	3	16
Egg and egg products	89	1	0	1	1	1	1	1

^1^ Mean = the average number of ingredients in products based on the on-pack statement of ingredients. All main and first-level sub-ingredients are included in the total count of ingredients. ^2^ SD = standard deviation.

## Data Availability

The data presented in this study are available on request from the corresponding author. The data are not publicly available due to privacy policies.

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
