# Peer review of "Deconstructing the Supermarket: Systematic Ingredient Disaggregation and the Association between Ingredient Usage and Product Health Indicators for 24,229 Australian Foods and Beverages"

_nutrients, 2021, doi:10.3390/nu13061882_

Round 1

Reviewer 1 Report

This subject is the great impact and useful in different fields. The authors should better mark the novelty character of the paper and the wide range of applications.

The authors should better mark in the introduction the importance of communication among food composition data and food consumption data and the procedures of standardization and harmonizations and related references should be mentioned such as:

Durazzo A., Lisciani S., Camilli E., Gabrielli P., Marconi S., Gambelli G., Aguzzi A., Lucarini M., Maiani G., Casale G., Marletta L. Nutritional composition and antioxidant properties of traditional Italian dishes. Food Chem. 2017  218:70-77. doi: 10.1016/j.foodchem.2016.08.120. 

Traka M. Maintaining and updating food composition datasets for multiple users and novel technologies: Current challenges from a UK perspective. Nutrition Bulletin 45(2):230-240. 

In Material and Methods, each step of procedure should be better described. Moreover some consideration of future direction of research concerning application of description and Classification system such as LanguaL and FoodEx2 should be inserted and related references.

Data in Table 1 should be better described in the text

Conclusion should be greatly implemented including limits, advantages and future applications

Author Response

Dear Sir/Madam,

Re Manuscript nutrients-1223305

Thanks for reviewing this report.  Please find below responses to comments attached.  We hope the paper will now be suitable for publication.

Kind regards,

Allison Gaines

Reviewer comments with inline replies:

This subject is the great impact and useful in different fields. The authors should better mark the novelty character of the paper and the wide range of applications.

The authors should better mark in the introduction the importance of communication among food composition data and food consumption data and the procedures of standardization and harmonizations and related references should be mentioned such as:

Durazzo A., Lisciani S., Camilli E., Gabrielli P., Marconi S., Gambelli G., Aguzzi A., Lucarini M., Maiani G., Casale G., Marletta L. Nutritional composition and antioxidant properties of traditional Italian dishes. Food Chem. 2017  218:70-77. doi: 10.1016/j.foodchem.2016.08.120. 

Traka M. Maintaining and updating food composition datasets for multiple users and novel technologies: Current challenges from a UK perspective. Nutrition Bulletin 45(2):230-240. 

  • We have added comment in the Introduction section regarding how this work might contribute to food composition analysis and be used to build upon existing food composition tables used in nutrition research. Related references have been added as suggested and including additional attribution to FSANZ standard and governed food composition tables. See page 2.
  • The two suggested references have been included in the Discussion section. See page 8.

In Material and Methods, each step of procedure should be better described.

  • We have included an additional figure to better explain the methods and steps taken in the analysis. See Appendix Figure A1.

Moreover some consideration of future direction of research concerning application of description and Classification system such as LanguaL and FoodEx2 should be inserted and related references.

Conclusion should be greatly implemented including limits, advantages and future applications

  • Commentary about the potential application to food chemistry and nutrition science research, as well as to sustainable diets research has been added in the Discussion section. See page 8.

Data in Table 1 should be better described in the text

  • Table 1 is described in two parts in the text, first on page 4 describing unique ingredients by major ingredient category; and second on page 5 describing ingredient usage in products. The caption further explains the terminology used in the column headings. We have made some refinements to the description to facilitate interpretation.  See pages 4-5.

Reviewer 2 Report

Report on “Deconstructing the supermarket: systematic ingredient disaggregation and the association between ingredient usage and product health indicators for 24,229 Australian foods and beverages” by Allison Gaines  et al.

The manuscript investigates the associations between the number of ingredients and indicators of product healthiness in Australian. I think the manuscript it is well written and quite clear. Most of the methodology is original and well designed, I have only one mayor concern:

As it can be seen in Figure 1 provided by authors, the distribution of the number of ingredients did not follow a normal distribution, it has a clear right asymmetry. In this case, the ANOVA test cannot be used to analyze the association between number of ingredients and the nutrient-based Health Star Rating (HSR) and the NOVA level-of-processing classification. You should use the non-parametric Kruskal-Wallis test to analyze the difference between groups.

I think this point must be corrected despite the boxplot in Figure 2 provided by authors it is the best way to analyze this association.

Author Response

Dear Sir/Madam,

Re Manuscript nutrients-1223305

Thanks for reviewing this report.  Please find below responses to comments attached.  We hope the paper will now be suitable for publication.

Kind regards,

Allison Gaines

Reviewer comments with inline replies:

Report on “Deconstructing the supermarket: systematic ingredient disaggregation and the association between ingredient usage and product health indicators for 24,229 Australian foods and beveragesby Allison Gaines et al.

The manuscript investigates the associations between the number of ingredients and indicators of product healthiness in Australian. I think the manuscript it is well written and quite clear. Most of the methodology is original and well designed, I have only one mayor concern:

As it can be seen in Figure 1 provided by authors, the distribution of the number of ingredients did not follow a normal distribution, it has a clear right asymmetry. In this case, the ANOVA test cannot be used to analyze the association between number of ingredients and the nutrient-based Health Star Rating (HSR) and the NOVA level-of-processing classification. You should use the non-parametric Kruskal-Wallis test to analyze the difference between groups.

I think this point must be corrected despite the boxplot in Figure 2 provided by authors it is the best way to analyze this association.

  • We have updated the analysis to be a Kruskal-Wallis H test with post-hoc Dunn pair-wise analyses as suggested. See page 7.
  • We have also updated all results to display median (IQR) instead of mean (SD). See changes throughout document.